# Non-destructive lock-picking of a historical treasure chest by means of X-ray computed tomography

**Eva Zikmundová[1,2], Tomáš Zikmund[1], Vladimír Sládek[3], Jozef Kaiser [1] \***

**1** Central European Institute of Technology, Brno University of Technology (CEITEC BUT), Brno, Czech Republic, **2** Department of Chemistry, Faculty of Science, Masaryk University, Brno, Czech Republic, **3** Department of Anthropology and Human Genetics, Faculty of Science, Charles University, Prague, Czech Republic

\* jozef.kaiser@ceitec.vutbr.cz

**Data Availability Statement:** The whole CT dataset underlying the findings of this study are available on the Open Science Framework repository at DOI: 10.17605/OSF.IO/NXPHY.

## Abstract

An innovative approach to a non-destructive lock mechanism examination by means of X-ray computed tomography (CT) was involved in a careful opening of a locked 19[th] century chest missing the key, as an interdisciplinary cooperation with the restorers. In regard of the exploration and conservation of such locked objects, their opening is important to the restorers. However, the opening may be complicated, if not impossible, without damaging the object when the key is missing. Moreover, the historical locks might be equipped with protective mechanisms. Despite the exceeding dimensions and the weight of the steel chest, a CT analysis was performed, which enabled a detailed exploration of the lock based on a system of levers and bolts handled by a single key, located in a case on the inside of the chest lid, including the dimensions essential for manufacturing of a new key copy. Moreover, two secret protective mechanisms were revealed, as well as all the damages of the object.

## Introduction

X-ray computed tomography (CT) [1] enables to investigate non-destructively the objects of cultural heritage. The technique is based on a sample scanning using the trans-illuminative X-ray radiation [2], and a 3D model can be obtained as a result [3]. The dense parts of the scanned material reduce the intensity of the radiation, as the X-rays are partly absorbed. The sample projections represented by different intensities for different materials of the sample are then detected and the data acquired over an angular range of 360° are mathematically processed into virtual cross-sections (CT data) [4] including the detailed information about the inner structure of the material up to a micro-level resolution [5]. Moreover, X-rays cannot cause any harm to inanimate objects (unlike biological samples) and the scanning is, therefore, non-invasive. These advantages make CT a promising technique in the study of the cultural heritage objects, such as various wooden artefacts [6–9] including musical instruments [10, 11], pottery [12] and skeletal remains [13]. However, the use of commercially available medical [14] or industrial [15] CT systems is limited due to the achieved resolution or the size of the

**Funding:** This research was carried out under the project CEITEC 2020 (LQ1601) with financial support from the Ministry of Education, Youth and Sports of the Czech Republic under the National Sustainability Programme II, CEITEC Nano Research Infrastructure (MEYS CR, 2016-2019) and Ceitec Nano+ project, CZ.02.01/0.0./.0.0./16_013/0001728 under the program OP RD. J.K. acknowledges the support of the Brno University of Technology through grant FSI-S-17-4506.

**Competing interests:** The authors have declared that no competing interests exist.

objects. Thus, special systems have been developed and used in the analysis of large objects, e. g. in Bologna and Turin, Italy [16, 17] or Ghent, Belgium [11]. In this study, an innovative approach to open a locked historical chest is suggested via a non-destructive exploration of the lock mechanism using CT measurements.

In museum depositories throughout the Czech Republic, there are many locked objects missing the key [18, 19]. Opening of these objects is essential not only for the restoration process but also for the exploration of the inner space and possible content, for the examination of the lock mechanism and its functionality, and eventually for a new key copy manufacturing. A presumably locked steel chest, currently in the property of the South Moravian Museum in Znojmo, Czech Republic, which might have served as a guild or a city treasure chest was used to demonstrate an innovative approach of the lock mechanism exploration by applying CT.

Historical chests were used as early as in the Ancient Egypt [20–22] with a wide range of purposes. With the establishment of guilds, chests were used as vaults to store important documents, valuable objects and also money of the guilds or of the city councils [23, 24]. However, they also served for the ceremonial purposes–the plenary guild meetings used to be initiated by opening of the chest and ended by closing it [25]. The sanctity of the open chest was significant, as the decisions made while the chest was opened were considered legitimate and binding. Disrespectful behaviour was even considered as a violation of the ceremony and was penalized [24]. The chests could also have a social function, when they were carried in a carnival procession during their transportation to a new location or as a part of celebrations and festivals [23]. With the abolition of the guilds in the 19<sup>th</sup> century, the guild chests lost their purpose and became the relics [23, 25].

The chests containing the valuables were highly treasured, therefore, several safety precautions were involved. The treasure chests were made of wood and armoured with steel strips or entirely made of steel. To increase the protection, the chest could have been fastened to the ground [26], e. g. through the holes in the bottom. Thus, it could not be moved without its opening and dismounting. Moreover, the lock mechanisms were used as a protection. The chests used to be equipped with more locks with different keys usually held by the guild masters or councilmen and it was only possible to open the chests when all the keys were present. Hidden keyholes and even false ones or lock bolts directed into three or four sides of the chest were among many devices often present in order to increase the protection of the chest's content [25].

The impossibility of the chest opening may have various causes–the key absence, the damage of the lock mechanism, the presence of the corrosion products, etc. Any destructive technique, including e. g. cutting the bottom of the chest, is, however, barely acceptable by the restorers and it is used only if inevitable.

A possibility of a non-destructive examination of the construction and potential defects of a steel treasure chest lock mechanism by means of CT is introduced in this work. We encountered several challenges, such as the very positioning of the chest in the CT cabinet regarding the large dimensions and weight of the object, a possible data distortion due to the thickness and a high attenuation character of the analysed material, etc. Despite them, the goal is to provide a detailed construction model with precise dimensions to be used for a new key copy manufacturing and for understanding the functionality of the lock mechanism.

## Materials and methods

### Treasure chest

The historical chest investigated in this study currently belongs to the collections of the South Moravian Museum in Znojmo, Czech Republic. Its origin is unknown, but the chest was dated

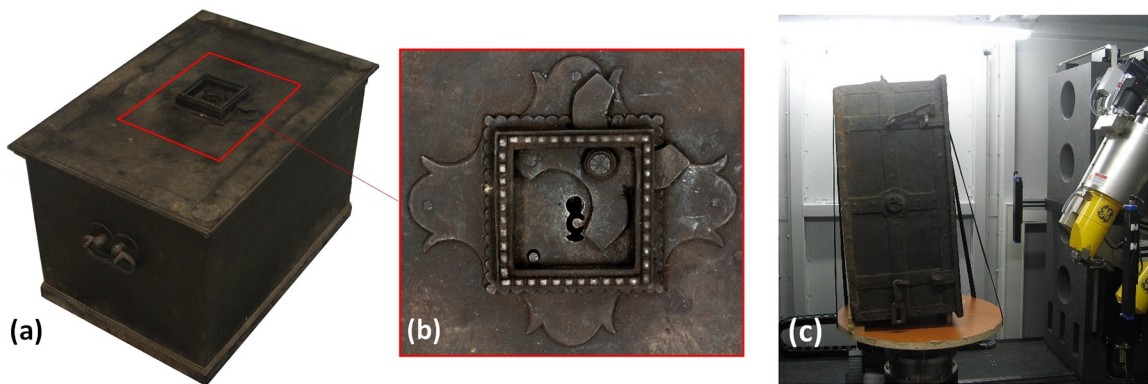

**Fig 1.** (a): Analysed chest; (b): detail of the lock [27] located in the centre of the chest lid; (c): illustration of a chest fixation in the CT system.

to the 1$^{st}$ half of the 19$^{th}$ century by the art historian PhDr. Jan Mohr [27]. It has the dimensions of 580 x 392 x 410 mm and because it is completely made of steel, its weight is 60 kg. The maximum material thickness of the scanned area is approximately 500 mm, representing the width of the lid, which was a challenge for the CT measurement. It seems plain, without decorations. In the bottom, there are four holes, probably enabling to fasten the chest to the ground. The original key is missing.

The lock mechanism is located in the centre of the lid (Fig 1(a) and 1(b)). The keyhole is framed in a scalloped square-shaped frame, underlaid with one similarly scallop-edged plate and a second bigger plate forming a simple floral motif. An X-shaped cap is attached to the frame as a protection of the keyhole.

## X-ray computed tomography

A CT measurement of the lock mechanism was performed using a GE phoenix v|tome|x L 240 industrial CT system equipped with a 240 kV/300W maximum power X-ray micro focus tube and a high- contrast flat-panel detector DXR250 with a 2048 × 2048 pixel, 200 × 200 µm pixel size [28]. The granite based 7-axis manipulator allows placing a bulky object (see Fig 1(c)). The exposure time was 850 ms in each of 1800 projections. The position of the detector for every X-ray image was randomly shifted during the acquisition process in order to eliminate the ring artefacts [29]. The microCT scan was carried out at the maximum possible acceleration voltage (240 kV) and a 270 µA X-ray tube current, i.e. a power of 64.8 W. The X-ray spectrum of a tungsten target was modified by 0.5 mm Cu and 0.5 mm Sn filters to reduce the beam hardening [30]. The tomographic measurement was performed at the temperature of 21 ˚C. The detector distance at 1255 mm and the object distance at 639 mm gave the magnification of 1.9 and the angle of a cone beam of 18˚. The isotropic linear voxel size of the obtained volume was 102 µm. This defined the field of view of 20 cm × 20 cm, which was focused on the lock area (Fig 2, the scanned area is marked in Fig 4), i.e. region of interest tomography [31]. The tomographic reconstruction was realized using the GE phoenix datos|x 2.0 3D computed tomography software [28] based on the filtered back projection algorithm [4]. Within this software, the object shifting correction and the beam hardening correction in a different material mode (number set to 8.5) was applied [32].

The VG Studio MAX 2.2 [33] software was used for all visualizations of the CT data and the measurement of the key dimensions. Individual parts of the lock mechanism were transformed into geometric objects which were subsequently colour-coded in a 3D visualization. The

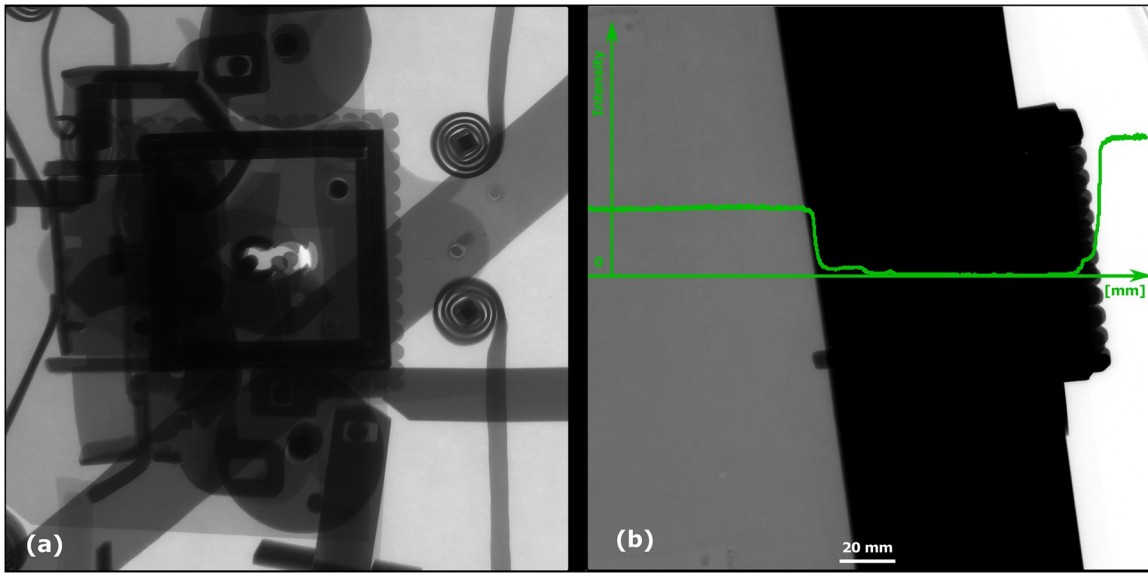

**Fig 2. Radiographic image in a sample rotation by (a) 0˚ and (b) 90˚ with no signal in the dark area; the intensity plot is pictured in green (the black colour is represented by the lowest intensity, the highest intensity corresponds to the white colour).**

transformation into geometric objects was done manually. Based on the edges of the lock mechanism parts, polygonal formations were drawn using VG Studio software.

## CT data

The chest is made of a high attenuation material, which caused the beam hardening and the scattering of the X-ray radiation [29]. This brought various artefacts into the images in the form of bright/dark streaks, variability in the intensities and the silhouettes from the neighbouring slices (see Fig 3(a)). Furthermore, the large dimensions of the chest lid (a projecting

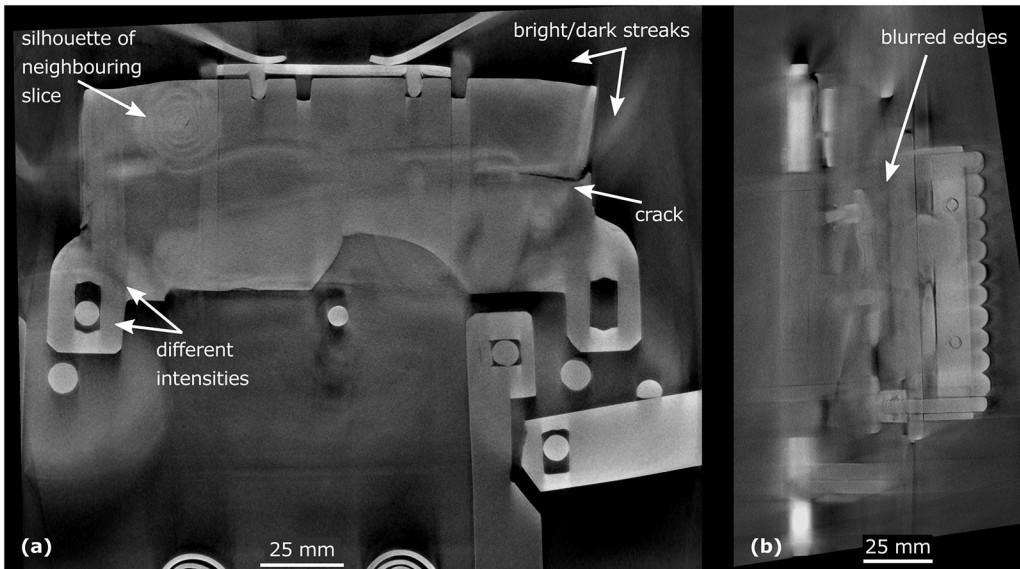

**Fig 3. Selected transverse (a) and longitudinal (b) cross-section.**

**Table 1. An overview of characteristic lock parts; the typical dimensions are marked red in the pictograms (l = length, w = width, th = thickness, Ø = diameter).**

| | Lock part | Pictogram | Typical dimension |
|---|---|---|---|
| | Scalloped frame |  | l: 92 mm w: 92 mm |
| | Case |  | l: 440 mm w: 285 mm th: 3 mm |
| **First mechanism** | Keyhole |  | l (key bit): 23.6 mm Ø: 11.5 mm |
| | Peg |  | l: 21 mm Ø: 6.2mm |
| | Catcher |  | l: 125 mm th: 1.5 mmw: 37 mm |
| | Arresting plate |  | l: 68 mm w: 65 mm detents distance: 14 mm th: 6 mm |
| | Bar |  | l: 152 mm w: 63 mm th: 6 mm |
| | Guiding wheels |  | th: 5 mm Ø: 64mm |
| | Bolts |  | w: 20 mm th: 7 mm |
| **Second mechanism** | Secret button |  | l: 15 mm th: 5 mm |
| | X-shaped cap |  | l: 81 mm th: 5 mm w: 77 mm peg distance from the rotation axis: 62 mm |
| | Peg |  | l: 13 mm Ø: 7 mm |
| | Bolts |  | w: 21 mm th: 3 mm |

(*Continued*)

**Table 1.** (Continued)

| | Lock part | Pictogram | Typical dimension |
|---|---|---|---|
| **Third mechanism** | Cradle | | button to axis distance: 40 mm th: 3.5 mm peg to axis distance: 28 mm |
| | Peg | | l: 18 mm Ø: 5 mm |

material thickness of about 500 mm in one direction) completely shielded the X-ray radiation in a few rotation positions around 90° (see Fig 2). That led to unclear or blurred edges of the steel sheets (see Fig 3(b)). All of these issues made an automatic segmentation impossible. Nevertheless, the CT images allowed detecting e. g. a 29 mm long crack in the bar (see Fig 3(a)) and they were pictured by enough contrast to distinguish the edges of each component.

## Results

### Lock mechanism exploration

Three different and mutually independent mechanisms, hidden in a case on the inside of the chest lid, were revealed by the CT analysis. An overview of the parts of the lock belonging to individual mechanisms with their typical dimensions is summed up in Table 1.

The first mechanism controlled by the key is a system of bolts and levers (Fig 4). It is necessary to insert a right-shaped key which would pass both through the keyhole and the second aperture (Fig 5(aI) and 5(aII)) respectively) of a slightly different shape hidden inside the lid. In the lock, the hollow key shank is slid onto a peg (Fig 5(aIII)) which serves as a rotation axis

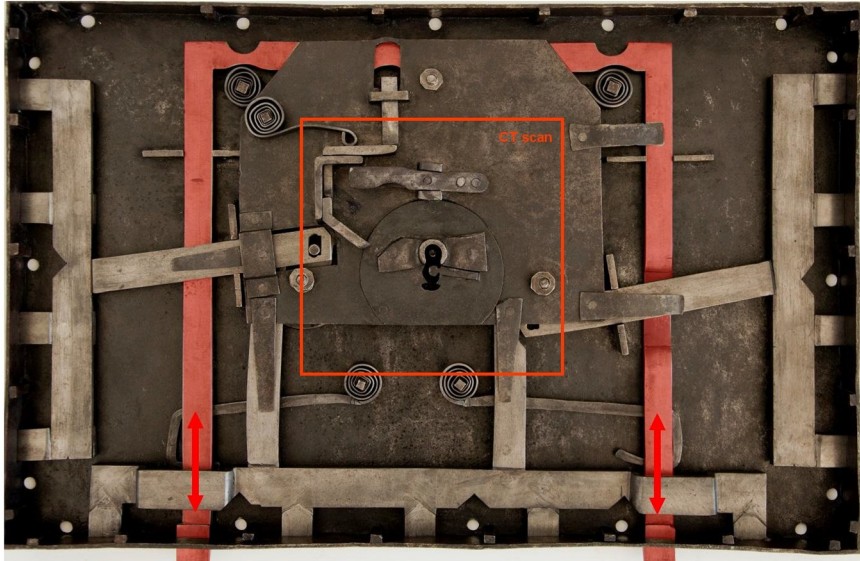

**Fig 4. Complete lock mechanism of the chest in the case from below the covering plate (after restoration)** [27] **including the bolts and levers of the first mechanism; the lever of the second mechanism with the bolts in a locked position is marked red with the direction of their motion indicated by the arrows.**

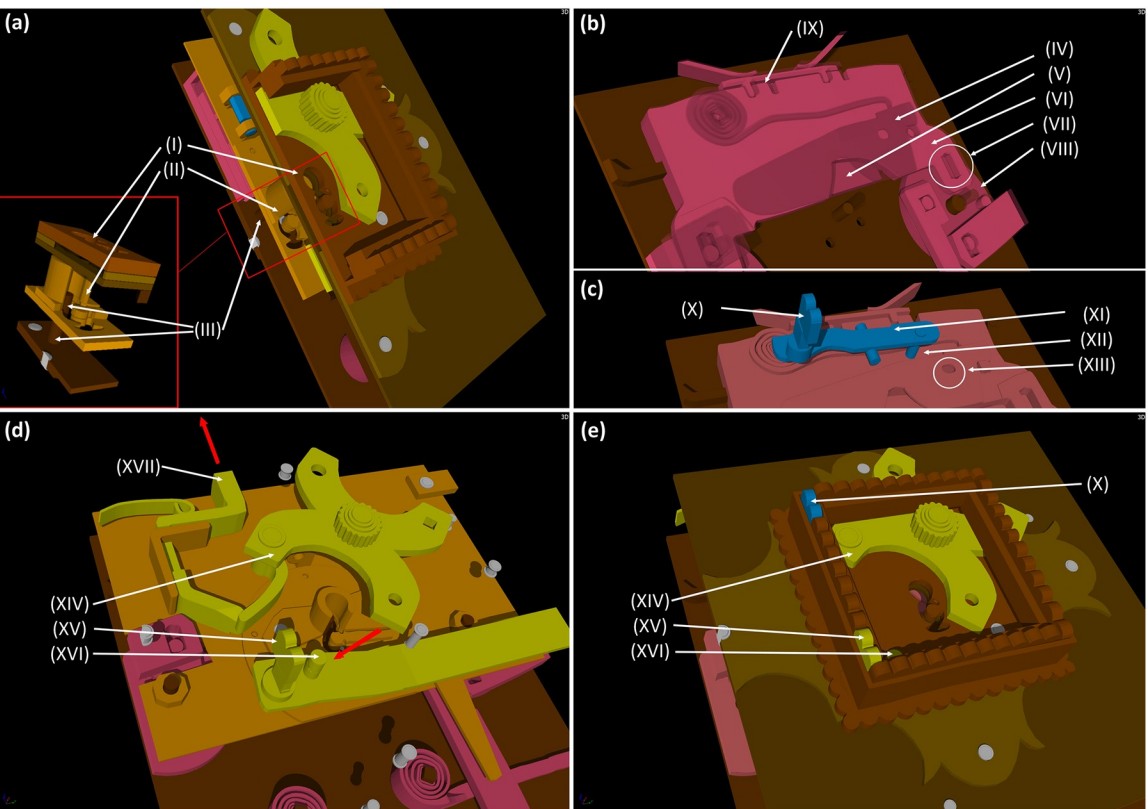

**Fig 5.** A 3D model of the lock mechanism from the CT data; (a): a cross section with a detail of the keyhole depicting: *(I)* the keyhole, *(II)* the second aperture of the lock and *(III)* the peg onto which the key shank is being slid; (b): first mechanism: *(IV)* the catcher, depicted semi-transparent, *(V)* the arresting plate, partly hidden by *(VI)* the bar, *(VII)* the missing peg that should connect the bar with the *(VIII)* guiding wheel, *(XI)* the detents of the bar with the arresting plate fit in the unlocked position; (c): parts of the third mechanism depicted in blue: *(X)* the secret button, *(XI)* the cradle, *(XII)* the peg, which was supposed to fit in *(XIII)* the hole to block the catcher; (d +e): parts of the second mechanism depicted in yellow as seen on the lid of the chest: *(XIV)* the X-shaped cap covering the keyhole, *(XV)* the secret button, *(XVI)* the peg supposed to fix the cap in the closed position, *(XVII)* a little lever transferring the motion of the X-shaped cap onto the main lever with the bolts (see Fig 4), direction of the motion to the opened position is marked by the red arrows, *(X)* the third mechanism secret button depicted in blue.

of the key. By turning the key, the catcher and the arresting plate (Fig 5(b*IV*) and 5(*V*) respectively) are lifted and the bar (Fig 5(b*VI*)) is shifted. This movement is transferred onto the levers with the bolts via two guiding wheels (Fig 5(b*VIII*)). Thirteen bolts on the three sides of the case are pushed in or out. When the key is fully turned, the arresting plate falls back into the detents of the bar (Fig 5(b*IX*)) which is held until the key is turned again.

The second mechanism is connected with an X-shaped keyhole cap (Fig 5(d) and 5(e*XIV*)) which can be hold with a peg (Fig 5(d) and 5(e*XV*)) in the lower left corner of the lock when turned to hide the keyhole. In order to reveal the keyhole, it is necessary to press a secret button (Fig 5(d) and 5(e*XVI*)) in the lower left corner of the frame hidden in the decoration. The mechanism connects the cap with the lever ended with two lock bolts on one side of the case (see Figs 4 and 5(d*XVII*)). The bolts prevent the lid from opening when the keyhole is revealed, and it is necessary to turn the cap back after unlocking.

The third mechanism is controlled by the second secret button hidden in the frame decoration (Fig 5(c) and 5(e*X*)). By pushing the button, a cradle (Fig 5(c*XI*)) supported by a spring from the board with the second keyhole is lifted. The cradle is ended with a peg (Fig 5(c*XII*)) reaching above the catcher. In the catcher, there is a hole (Fig 5(c*XIII*)) probably for the peg to

fit in and fix it. This mechanism was probably meant to block the catcher under particular circumstances. The peg was supposed to fit in the hole during unlocking and prevent any further movement of the first mechanism. In such case, it would be necessary to push the secret button when turning the key.

## Discussion

### Chest opening and the obstacles

Based on the 3D model, it was found out that the mechanism had actually been unlocked during the analysis. The first mechanism was partly broken, as the connection of one of the guiding wheels and the bar was missing (Fig 5(b*VII*)). However, this was not an obstacle for the chest opening thanks to a system of levers outside of the scanned area connecting both wheels that transferred the movement of one wheel to the other one. It was not possible to repair this malfunction, probably caused during the manufacturing of the chest, as the parts of the guiding mechanism would have to be rearranged.

The second mechanism was not functional either because it was stuck in the closed position regardless of the cap movement. It was estimated and consequently confirmed that there had been a misplaced spring outside of the scanned area which usually served to push the lever with the bolts back. The misplaced spring was actually the reason why the chest could not be opened.

The intended function of the third mechanism was explained neither by the CT analysis nor after dismantling of the lock during the restoration intervention. It was not possible to lift the catcher enough by turning the key, thus, the trajectory of the hole could not reach the position of the peg in any way (Fig 6). A hypothesis was formed that the mechanism had never

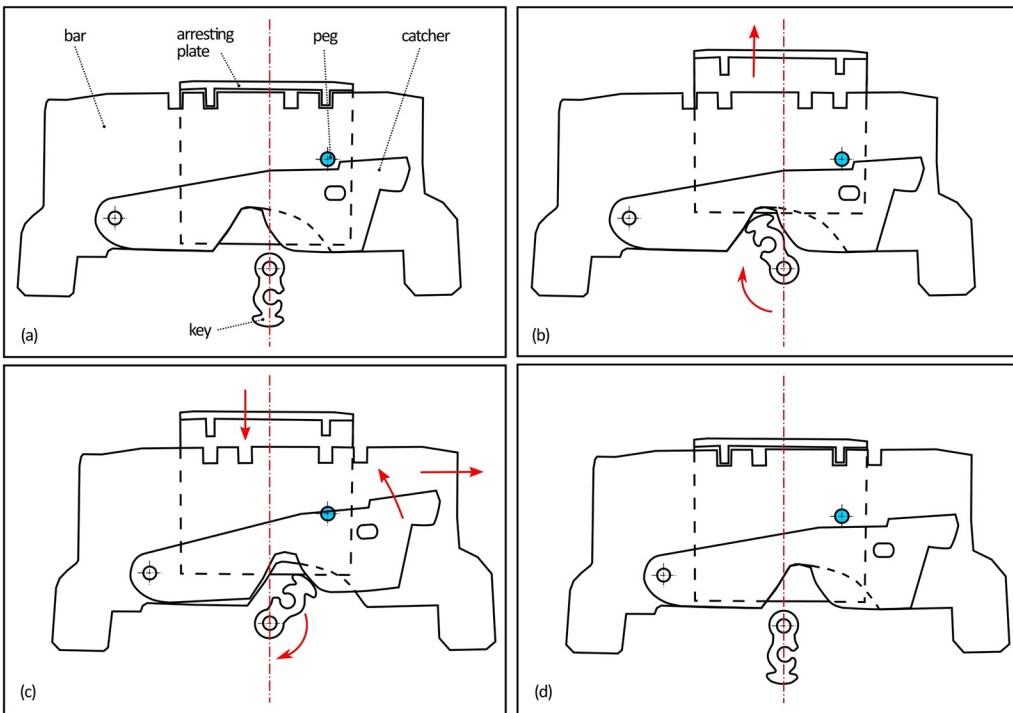

**Fig 6. The first mechanism motion scheme from the locked position *(a)* to the unlocked position *(d)* marked by the arrows, demonstrating the malfunction of the third mechanism; the trajectory of the hole cannot reach the position of the peg (marked blue).**

actually been functional, because no signs of damage were indicated, and it was still possible to open the chest despite the non-functionality of the third mechanism.

If the key was not missing and all the mechanisms were fully functional, three steps would be necessary in the opening process. The first secret button (lower left corner of the frame) would have to be pushed to turn the X-shaped cap and would reveal the keyhole. The second step would be inserting the key, pushing the second secret button (upper right corner of the frame) and turning the key. The final step, removing the key and turning the X-shaped cap back to hide the keyhole, would follow. Only after these three steps it would be possible to open the chest.

## Restoration intervention

A temporary key copy was created based on the shape and dimensions (Table 1) acquired from the CT measurement.

The only repairable issue preventing the opening was the loosened spring of the first mechanism. The lock bolt connected to the spring had to be pushed by a long custom-made iron rod inserted into the chest through the hole in its bottom, possibly used to fasten the chest to the ground. After opening the chest, it was found out that the spring had only fallen out from its position and it was possible to repair the mechanism immediately.

After the opening, the whole chest including the lock mechanism was dismantled, there was no content hidden inside. The material of the chest suffered from a corrosion attack and it was polluted by dust and grease. Therefore, all the parts of the chest were cleaned and preserved. An original dark green surface coating in a very good condition was revealed. The corrosion products were removed or stabilized. Several missing minor parts, such as nuts or pegs, were replaced if possible.

The lock mechanism was repaired and it was possible to lock it again. Despite the functionality of the mechanism, it was recommended by the restorer not to lock the chest, because the parts of the mechanisms were worn out and they could be easily broken down again. Such a damage would probably lead to a necessary re-conservation of some of the parts or it could even prevent the chest opening again. As the temporary key copy used to open the chest before the intervention was not esthetical enough, a second copy (see Fig 7) was manufactured for the exhibition purposes [27].

## Conclusions

The non-destructive exploration of the 19th century treasure chest lock mechanism by means of X-ray computed tomography was carried out successfully. Despite several restraints, such as the high absorbing material and the size of the chest, the obtained tomographic data were of a sufficient quality and they allowed creating a 3D model of the lock mechanism. Therefore, it was possible to explore all parts of the lock in detail, to reveal the potential damages and to obtain the dimensions required for a new key copy. It was found out that the chest had not been locked and the only issue preventing it from opening was, in fact, a misplaced spring of the second mechanism. Based on the acquired information, it was possible to open the chest non-destructively. A conservation and restoration intervention was performed including the lock mechanism repair and the key manufacturing. Since any destructive approach is always preferably avoided by the restorers, this method might become a useful tool in the cultural heritage preservation practice.

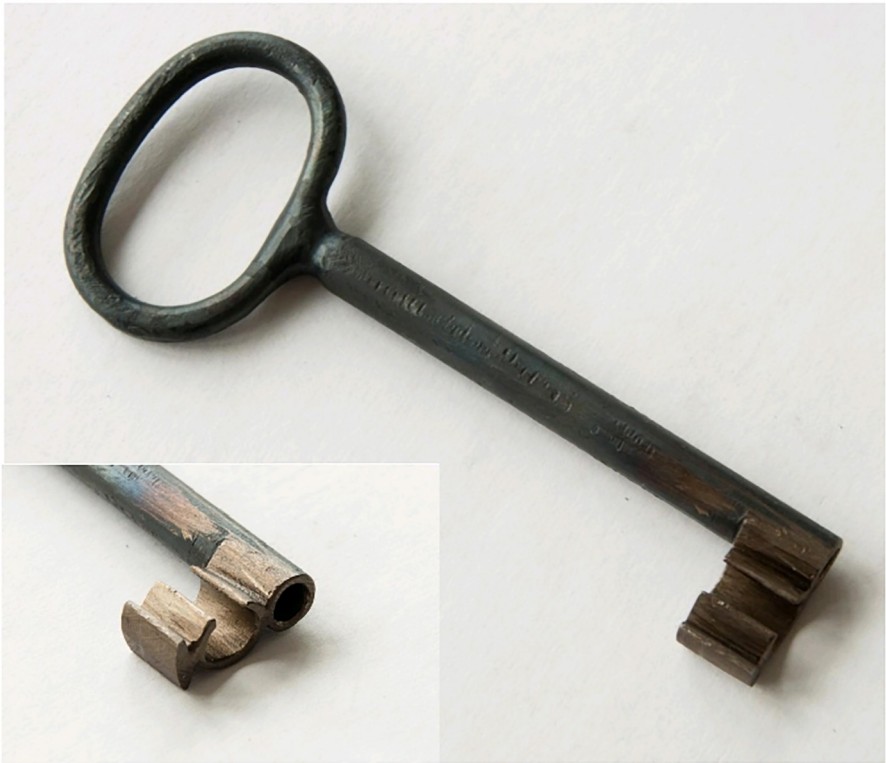

**Fig 7. Second manufactured key copy; the key bit is pictured in two different angles.**

## Supporting information

**S1 File. Lock mechanism of a historical chest.**
(PDF)

## Author Contributions

**Conceptualization:** Jozef Kaiser.

**Data curation:** Eva Zikmundová, Vladimír Sládek, Jozef Kaiser.

**Formal analysis:** Eva Zikmundová, Vladimír Sládek.

**Funding acquisition:** Jozef Kaiser.

**Investigation:** Tomáš Zikmund.

**Methodology:** Tomáš Zikmund.

**Project administration:** Jozef Kaiser.

**Resources:** Jozef Kaiser.

**Supervision:** Jozef Kaiser.

**Validation:** Eva Zikmundová, Vladimír Sládek, Jozef Kaiser.

**Visualization:** Eva Zikmundová, Tomáš Zikmund.

**Writing – original draft:** Eva Zikmundová, Tomáš Zikmund, Vladimír Sládek, Jozef Kaiser.

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
