## [Decision Letter · Decision Letter 0]

18 Feb 2020

PONE-D-19-30241

Non-destructive lock-picking of a historical treasure chest by means of X-ray computed tomography

PLOS ONE

Dear Prof. Kaiser,

Thank you for submitting your manuscript to PLOS ONE. After careful consideration, we feel that it has merit but does not fully meet PLOS ONE’s publication criteria as it currently stands. Therefore, we invite you to submit a revised version of the manuscript that addresses the points raised during the review process.

In response to the reviewer's suggestion that the English in the paper should be improved, I suggest you thoroughly copyedit your manuscript for language usage, spelling, and grammar. If you do not know anyone who can help you do this, you may wish to consider employing a professional scientific editing service. Whilst you may use any professional scientific editing service of your choice, PLOS has partnered with American Journal Experts (AJE) who have experience helping authors meet PLOS guidelines and can provide language editing, translation, manuscript formatting, and figure formatting to ensure your manuscript meets our submission guidelines. To take advantage of our partnership with AJE, visit the AJE website (http://learn.aje.com/plos/) for a 15% discount off AJE services.

Upon resubmission, please provide the following: · The name of the colleague or the details of the professional service that edited your manuscript

· A copy of your manuscript showing your changes by either highlighting them or using track changes

(uploaded as a *supporting information* file)

· A clean copy of the edited manuscript (uploaded as the new *manuscript* file)

I also note from your data availability statement that the underlying data is currently available on request. PLOS journals require authors to make all data underlying the findings described in their manuscript fully available without restriction, with rare exception. Please update your Data Availability Statement to indicate whether you will be able to make your data available at the time of acceptance and provide details of where the underlying dataset can be found. We only require you to provide the minimal dataset i.e. the data used to reach the conclusions drawn in the manuscript with related metadata and methods, and any additional data required to replicate the reported study findings in their entirety.  

We would appreciate receiving your revised manuscript by Apr 02 2020 11:59PM. To enhance the reproducibility of your results, we recommend that if applicable you deposit your laboratory protocols in protocols.io, where a protocol can be assigned its own identifier (DOI) such that it can be cited independently in the future. For instructions see: http://journals.plos.org/plosone/s/submission-guidelines#loc-laboratory-protocols

We look forward to receiving your revised manuscript.

Kind regards,

Dr Joseph Donlan

Staff Editor

PLOS ONE

Journal Requirements:

Reviewers' comments:

Reviewer's Responses to Questions

**Comments to the Author**

1. Is the manuscript technically sound, and do the data support the conclusions?

Reviewer #1: Yes

Reviewer #2: Yes

Reviewer #3: Yes

2. Has the statistical analysis been performed appropriately and rigorously? 

Reviewer #1: N/A

Reviewer #2: Yes

Reviewer #3: N/A

3. Have the authors made all data underlying the findings in their manuscript fully available?

Reviewer #1: No

Reviewer #2: Yes

Reviewer #3: Yes

4. Is the manuscript presented in an intelligible fashion and written in standard English?

Reviewer #1: No

Reviewer #2: Yes

Reviewer #3: Yes

5. Review Comments to the Author

Reviewer #1: I like this interesting and curious paper and support its publication. The explanation of the work is clear, but it is unfortunately let down by poor written English throughout the paper. I feel that the paper needs to be thoroughly proof-read before it can be published.

It is not clear to me how figure 5 was generated. Presumably it was made with some 3D modelling software but was it made by hand after looking carefully at the CT scan or generated directly from CT?

Please check PLOS's open data requirements. You say that the data are available from the author, but PLOS ONE might insist that you upload them to a repository.

Reviewer #2: This paper reflects the results of an X-ray CT study for a very bulky object, a wooden box (chest) with a quite complicated locking device, which should be opened without damaging the structure after understanding the mechanism. The object has some cultural heritage value and the key is missing. In this respect, the work is an interesting collaboration of CT experts with persons from the restoration and museum side.

The paper is well written and structured and gives most of the details in order to understand the object’s structure and composition and the applied non-invasive technique, including the image processing steps.

The challenge of the study is the large size of the object (bigger than the field-of-view or beam diameter) and the thickness of the material layers to be penetrated. In this respect, the study is cutting-edge and the limitations and problems are highlighted accordingly.

Despite of the good quality of the submitted draft of the paper, there are some aspects to improve the manuscript further:

1. Abstract: this study is by far not “a new approach” (since CT is available since decades), but a nice demo for using the method at the limit of the beam penetration. Further, the application to cultural heritage objects and the interaction with museums people could be highlighted more.

2. More details should be given in the material and method description: which materials are involved in detail (Fe, Cu, wood, …); Why 1800 single projections were produced? What are the X-ray beam parameters (cone, focal spot, beam geometry, distances of source, sample, detector, …)? What is the performance of the reconstruction method?

3. Table 1: please, add also the thickness of the individual components of the lock!

4. Furthermore, it would be interesting to compare extracted virtual parts with the real parts of the lock – this describes the quality of the study.

5. Fig. 7 shows only the performance of the workshop … better show the digital layout derived from the CT volume data … and compare to reality.

6. Fig. 6 doesn’t give a global overview of the lock and its parts. A comparison of Fig. 4 with the “virtual reality” should be given. Better views and maybe dedicated slices should be added and explained. Otherwise, the study remains incomplete.

7. From the methodical point, it could be added, that the study was performed in a regime where the sample is larger than the beam as a FOV tomography run. Because of the high number of taken projections, the missing information by the highly attenuating range seems to be overcome?

Reviewer #3: The paper describes an interesting application of X-ray Computed Tomography to the non-destructive examination of the lock mechanism of a 19th century chest with missing key. The aim was to obtain a precise and detailed model of the lock mechanism in order to understand its functionality and to manufacture a new key copy for opening the chest without any damage. 3D-CT is nowadays a well-consolidated technique also in the field of Cultural Heritage and it has been applied in a lot of case-studies. However, the present study is quite new and interesting because of the nature of the analyzed object and the high radiopacity of its constituent material that typically may produce artifacts in the reconstructed images decreasing their quality. In this case the authors were able to obtain tomographic images of good quality, so to characterize completely the lock mechanism.

The paper is well written and the results are original, so I suggest to accept the manuscripts with minor revisions.

In the following, my observations and suggestions.

Introduction

Line 31: I suggest writing “The sample projections…..are….” instead of “The sample projection …is...” and to specify that the data are acquired over an angular range of 360 degrees.

Line 40: Unlike the Reference 17, the CT system used for the analysis mentioned in Reference 16 was developed by a research group in Bologna, not in Turin.

Materials and methods

Line 104: I suggest using “projections” instead of “positions”

Line 108: In my opinion, Reference 31 is not entirely relevant, as it does not report the use of metal filters to reduce scattering artefacts, as stated by the authors. Moreover, to my knowledge, metal filters are generally used to reduce beam hardening, while collimators are more useful to reduce scattered radiation.

Lines 112-113: The sentence “Within this software…..was applied [32]” needs further explanations. In particular, what does it mean “…..in a different material mode (number 8.5)….”?

Line 127: Fig. 4(a) should be replaced by Fig. 3(a)

References

Line 300: the word “editors” should be deleted

Line 328: Insert “;” between “organizace” and “2014”

Line 331: Insert “;” between “techniques” and “2012”.

Finally, I suggest that the authors include (if possible) the data from the CT scan of the treasure chest as supplementary information.

6. PLOS authors have the option to publish the peer review history of their article (what does this mean?). If published, this will include your full peer review and any attached files.

Reviewer #1: No

Reviewer #2: No

Reviewer #3: No

---

## [Author Response · Author response to Decision Letter 0]

15 Apr 2020

Dear Editor,

Please find attached our revised manuscript entitled “Non-destructive lock-picking of a historical treasure chest by means of X-ray computed tomography” to you to consider for publication.

We revised the manuscript according the referee’s comments. We had the manuscript proof-read by Martina Pořízková, née Horníčková. Her University Diploma in English is attached as a supplementary material.

We added to a manuscript a Suplementary file a 3D pdf visualizing the Lock mechanism.

Regarding your remark

I also note from your data availability statement that the underlying data is currently available on request. PLOS journals require authors to make all data underlying the findings described in their manuscript fully available without restriction, with rare exception. Please update your Data Availability Statement to indicate whether you will be able to make your data available at the time of acceptance and provide details of where the underlying dataset can be found. We only require you to provide the minimal dataset i.e. the data used to reach the conclusions drawn in the manuscript with related metadata and methods, and any additional data required to replicate the reported study findings in their entirety.

we prepared a whole CT dataset underlying the findings presented in the manuscript to make fully available without any restrictions. The CT dataset (8bit stacks of tomographic slices in European coordinate system) are available at Open Science Framework, DOI 10.17605/OSF.IO/NXPHY

If you have any questions, please, do not hesitate to email (jozef.kaiser@ceitec.vutbr.cz) or call me (+420 731 141 281).

Thank you for your time spent on our manuscript.

Sincerely

Prof. Ing. Jozef Kaiser, Ph.D. corresponding author

---

## [Decision Letter · Decision Letter 1]

15 Jun 2020

Non-destructive lock-picking of a historical treasure chest by means of X-ray computed tomography

PONE-D-19-30241R1

Dear Dr. Kaiser,

We’re pleased to inform you that your manuscript has been judged scientifically suitable for publication and will be formally accepted for publication once it meets all outstanding technical requirements.

Kind regards,

Joseph P. R. O. Orgel, Ph.D.

Section Editor

PLOS ONE

Additional Editor Comments (optional):

Reviewers' comments:

Reviewer's Responses to Questions

**Comments to the Author**

1. If the authors have adequately addressed your comments raised in a previous round of review and you feel that this manuscript is now acceptable for publication, you may indicate that here to bypass the “Comments to the Author” section, enter your conflict of interest statement in the “Confidential to Editor” section, and submit your "Accept" recommendation.

Reviewer #1: All comments have been addressed

Reviewer #2: (No Response)

2. Is the manuscript technically sound, and do the data support the conclusions?

Reviewer #1: (No Response)

Reviewer #2: Yes

3. Has the statistical analysis been performed appropriately and rigorously? 

Reviewer #1: (No Response)

Reviewer #2: Yes

4. Have the authors made all data underlying the findings in their manuscript fully available?

Reviewer #1: (No Response)

Reviewer #2: Yes

5. Is the manuscript presented in an intelligible fashion and written in standard English?

Reviewer #1: (No Response)

Reviewer #2: Yes

6. Review Comments to the Author

Reviewer #1: All concerns have been adequately addressed.

Reviewer #2: The authors accepted the reviewers comments and improved accordingly. Details of the applied technique and the explanation of the function of the object are given much better.

7. PLOS authors have the option to publish the peer review history of their article (what does this mean?). If published, this will include your full peer review and any attached files.

Reviewer #1: No

Reviewer #2: Yes: Eberhard H. Lehmann, PSI, CH

---

## [Editor Report · Acceptance letter]

23 Jun 2020

PONE-D-19-30241R1 

Non-destructive lock-picking of a historical treasure chest by means of X-ray computed tomography 

Dear Dr. Kaiser:

I'm pleased to inform you that your manuscript has been deemed suitable for publication in PLOS ONE. Congratulations! Your manuscript is now with our production department. 

Kind regards, 

on behalf of

Prof. Joseph P. R. O. Orgel 

Section Editor

PLOS ONE